# Four New Cases of Progressive Ataxia and Palatal Tremor (PAPT) and a Literature Review

Norbert Silimon [1,*] , Roland Wiest [2] and Claudio L. A. Bassetti [1]

1 Department of Neurology, Inselspital, Bern University Hospital, University of Bern, 3010 Bern, Switzerland
2 Department of Diagnostic and Interventional Neuroradiology, Inselspital, Bern University Hospital, University of Bern, 3010 Bern, Switzerland
* Correspondence: norbert.silimon@insel.ch

**Abstract:** PAPT syndrome is a rare neurologic disorder characterized by progressive ataxia and palatal tremor (rhythmic movements of the soft palate). The first large study of PAPT patients was published in 2004, included a total of 28 sporadic PAPT cases, and suggested a neurodegenerative origin. In the last several years, case reports and small case series followed, underlining the heterogeneity of the clinical picture and underlying aetiology (including neurodegenerative, vascular, infectious/autoimmune, and genetic). As a contribution to the literature, we report on four new patients with PAPT syndrome from Bern. Our study highlights the diverse clinical presentation (pyramidal, extrapyramidal, bulbar, cognitive, psychiatric symptoms, and autonomic features), summarizes the known literature, and extends it by findings on sleep studies (obstructive/central sleep apnoea, sleep disturbance). Possible aetiologies and management aspects are discussed in light of the current literature.

**Keywords:** progressive ataxia and palatal tremor (PAPT); palatal tremor (PT); neurodegenerative; neurosarcoidosis; POLG; SCA20; HSP7; adult-onset Alexander's disease; NBIA; late-onset GM2-gangliosidosis; cerebrotendinous xanthomatosis

## 1. Introduction

Palatal tremor (PT), formerly also known as palatal myoclonus, is a rare movement disorder characterized by rhythmic movements of the soft palate [1]. The essential form—without known structural brain abnormalities—features isolated rhythmic contractions of the tensor veli palatini muscles, often leading to audible "ear click" sounds but no other neurological symptoms [2]. In the symptomatic form, a lesion in the pathways of the Guillain–Mollaret triangle (constituted by the ipsilateral inferior olive and red nucleus and the contralateral dentate nucleus) can lead to rhythmic contractions of the levator veli palatini muscles, as well as a plethora of other brainstem and cerebellar symptoms [3,4]. The most common lesions are vascular in aetiology (haemorrhagic and ischemic stroke, vascular malformations, and cavernomas), but brain trauma, brainstem tumours, and inflammatory/autoimmune aetiologies are described in the literature [5].

The hallmark of a third group of PT patients is associated with progressive ataxia [6]. This syndrome of progressive ataxia and palatal tremor (PAPT) is rare. The first mention of this syndrome was in 1985 [7], but the earliest case descriptions date back to 1954 [8].

In a first large study in 2004, an aggregate of 39 cases were reported (6 own cases; 22 sporadic, and 11 familial PAPT syndromes from the literature). That study was the first to summarize the phenotypic spectrum and categorize PAPT syndromes into sporadic and familial, thus bringing some order to the heterogeneity of case reports [6].

For many sporadic PAPT cases, a neurodegenerative aetiology is postulated [6,9–11]; however, some patients with structural brain lesions or autoimmune diseases were reported to develop PAPT as well [12–16]. Furthermore, rare genetic mutations can manifest with PAPT syndrome (familial PAPT), amongst other symptoms [17–22]. To our knowledge,

up until this point, 96 PAPT cases have been published in the English-speaking literature (49 sporadic/degenerative, 14 symptomatic, and 33 familial/genetic).

Here, we report the challenging cases of four patients with PAPT syndrome who were treated in the Neurology Department of the Inselspital, University of Bern, Switzerland, between 2014 and 2022.

Our centre serves a population of about 1–1.5 million persons as the only tertiary neurology department in the region. About 32.000–46.000 patients/year are seen in the outpatient clinics and about 3.200 patients/year are hospitalized. Consequently, we estimate the prevalence of PAPT between 0.26 and 1.05 per 100.000 in this region.

Our aim is to extend the known clinical and radiological characterisation of the syndrome, summarize known clinical and radiologic features, discuss possible aetiologies in light of the current literature review, and suggest diagnostic approaches.

## 2. Methods

We performed a systematic search of the digital medical records of patients in our hospital containing the keywords "palatal tremor OR palatal myoclonus" between 2014 and 2022.

We included patients having PT with progressive ataxia and excluded patients suffering from essential PT (1 patient) and symptomatic PT without progressive ataxia (7 patients). The final cohort consisted of 4 patients. Informed consent was obtained from all patients involved in the study. For the literature review, a thorough search of English-speaking literature was conducted in Pubmed using the keywords "palatal, tremor, myoclonus, ataxia, hypertrophic, olivary, degeneration" connected by the OR operator.

Of the PAPT syndromes found in the literature (neurodegenerative/structural/genetic), we calculated the mean age at onset with the standard deviation, gender proportions, and the percentage of imaging or pathologically proven cerebellar atrophy and HOD. The groups were compared by applying the appropriate statistical methods (the *p*-value was set at 0.05).

## 3. Patients Reports

### 3.1. Patient 1

In his early 40s, about 15 years prior, this male patient slowly developed gait ataxia, directional and pendular nystagmus, a partial horizontal gaze palsy, and dysarthria. An MRI of the brain showed atrophy of the cerebellum and T2-hyperintense lesions in the pons and the medulla oblongata. In the following aetiological workup, the patient showed multiple round foci in the CT of the chest. Due to an elevated ratio of CD4/CD8 positive cells in the bronchoalveolar lavage (the ratio was 11), sarcoidosis with neurological manifestations was postulated; unfortunately, however, a biopsy of the pulmonary findings was not obtained at that time. An immunosuppressive therapy with prednisolone and azathioprine was started without improvement of the neurological symptoms over 1.5 years. Two years later, a mediastinal lymphoma was diagnosed and successfully treated with chemotherapy and rituximab (no relapse over a 15 year follow-up period). The prednisolone therapy was continued for another 3 years after the chemotherapy without any effect on neurologic progression.

In subsequent examinations, a low-frequency palatal tremor (2–3 Hz) with synchronous myocloni of the lateral neck (no ear clicks) was detected. In addition, the patient developed slight asymmetric spasticity of the lower extremities. After a disease course of 15 years, the patient developed slight asymmetric limb ataxia and could walk with the aid of a cane. The neurocognitive examination was normal.

The examination of the cerebrospinal fluid (CSF) revealed a slightly elevated CSF–serum–albumin-index. Follow-up MRI examinations of the brain showed progressive atrophy of the cerebellum with persisting T2 signal alterations of the cerebellar peduncles and pons. The lesion in the dorsomedial pons was hypointense on susceptibility weighted images (SWI), corresponding to a small cavernoma or capillary telangiectasia and was

stable over 10 years. Eight years after symptom onset, sequential, bilateral hypertrophic olivary degeneration (HOD) was detected (Figure 1(1A–1D)). An MRI of the spine showed no signs of myelopathy or abnormalities of the spinal cord.

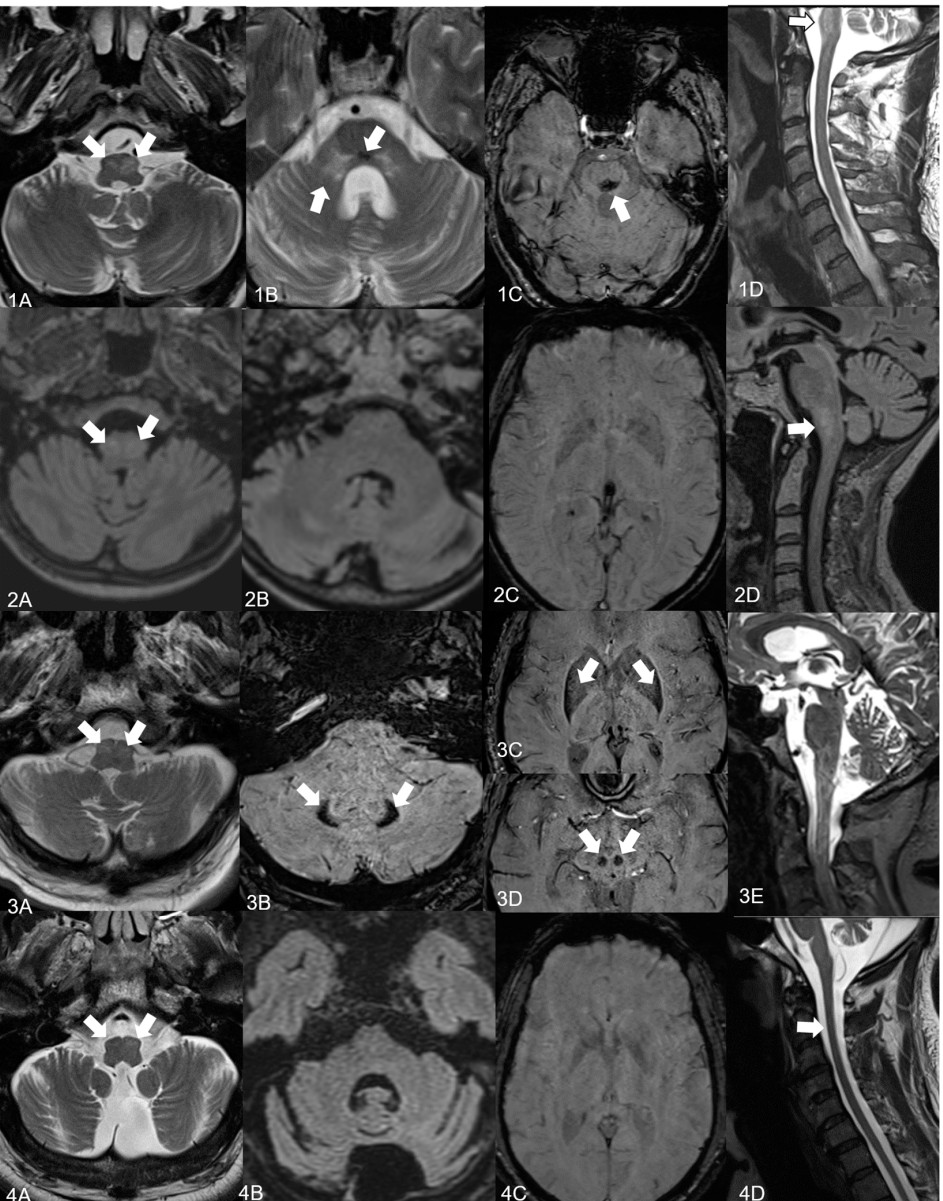

**Figure 1.** MRI of the brain and cervical spine. Patient 1: (**1A**,**1B**) T2 ax.: HOD left > right (arrows). Extensive T2 hypersignal in the medial cerebellar peduncles and a T2 hypointense lesion in the dorsomedial pons (arrows), (**1C**) SWI: Susceptibility artefact in the dorsomedial pons (arrow), (**1D**) T2 sag.: HOD, differential diagnosis: cavernoma or capillary teleangiectasia Patient 2: (**2A**) T2-FLAIR ax.: bilateral HOD (arrows), (**2B**) cerebellar atrophy (**2C**) SWI: no susceptibility artefacts, (**2D**) T2 sag.: marked HOD and cerebellar atrophy (arrow). Patient 3: (**3A**) T2 ax.: bilateral HOD (arrows), (**3B**) SWI: bilateral susceptibility artefacts in the dentate nucleus (arrows), (**3C**) the posterior and lateral putamen (arrows), (**3D**) and red nucleus (arrows), (**3E**) T2 sag.: cerebellar atrophy of both hemispheres and vermis. Patient 4: (**4A**,**4B**) T2 ax.: bihemispheric cerebellar and peduncular atrophy, bilateral HOD (arrows) (**4C**) SWI: no susceptibility artefacts (**4D**) T2 sag.: marked brainstem and cervical atrophy of the upper spinal cord (arrows). Abbreviations: ax.: axial plane, sag.: sagittal plane, FLAIR: fluid-attenuated inversion recovery; HOD: hypertrophic olivary degeneration, predominantly visible on T2w imaging as hyperintense signal abnormalities; SWI: susceptibility weighted image without focal inhomogeneities.

No sleep-related complaints or fatigue were reported (Epworth sleepiness scale (ESS) 1/24 points). On video-polysomnography (V-PSG), the patient only exhibited slight periodic limb movements, but no rapid eye movement sleep behaviour disorder (RBD) or sleep-associated breathing disturbance (Table 1). Genetic testing was suggested; however, the patient declined.

**Table 1.** Summary of the main findings in our PAPT patients.

| Patient | 1 | 2 | 3 | 4 |
|---|---|---|---|---|
| sex/AAO/DD | m/40/16 | f/42/26 | m/65/9 | m/50/4 |
| family history | − | − | − | − |
| first symptom | gait ataxia | gait disturbances | gait ataxia | gait disturbances, dysarthria, dysphagia |
| PAPT | + | + | + | + |
| PT with ear clicks | − | − | + | − |
| oculomotor symptoms | directional/pendular nystagmus horizontal gaze palsy | fixation deficit macro-square-wave-jerks dysmetric saccades | saccadic eye movement | hypometric saccades rotatory nystagmus |
| dysarthria/dysphagia | +/− | +/+ | +/+ | +/+ |
| spasticity | lower extremities | − | elevated reflex levels | spastic paresis of legs |
| polyneuropathy | − | + | − | + |
| hearing loss | − | + | − | − |
| autonomic dysregulation | − | − | − | + |
| cognitive symptoms | no impairment | 24 points (MoCA) | 23 points (MoCA) | 18 points (MoCA) |
| further neurologic symptoms | myocloni of the lateral neck fine motor disturbances | − | arm swing reduced | facial myoclonus hallucinations |
| medical history | possible sarcoidosis with neurological manifestations, mediastinal lymphoma | DM type 2, hypothyroidism, depression fatigue, daytime sleepiness | abdominal aortic aneurysm, extensive nicotine consumption | latent pulmonary tuberculosis |
| MRI cerebellar atrophy brainstem atrophy supratentorial atrophy HOD left/right further findings | + − − +/+ (sequential) SWI lesion in dorsomedial pons T2 signal alterations of the cerebellar peduncles | + − − +/+ | + − + +/+ susceptibility artefacts in the posterior and lateral putamen, red nucleus, dentate nucleus on SWI | + + + +/+ (sequential) |
| Serum abnormalities | − | − | elevated ferritin level | gliadin Ig A |
| CSF abnormalities | slightly elevated liquor–serum–albumin index | − | − | slightly elevated cell count (5–8 cells/μL) and protein |
| genetic testing | − | − | − | negative |

**Table 1.** *Cont.*

| Patient | 1 | 2 | 3 | 4 |
|---|---|---|---|---|
| PSG | AHI 5.7/h, ESS 1/24p. slight PLM no RBD no stridor | AHI 60.3/h, ESS 12/24p, FSS 63/63p. severe obstructive sleep apnoea fragmentation of sleep without REM sleep | AHI 21.1/h (Apnoe-Link) PSG was refused by patient ESS 6/24p. | AHI 71.6/h, ESS 10/24p. reduced sleep latency severe obstructive sleep apnoea, hypercapnia (mean $CO_2$ 54 mmHg) PLM, no RBD |
| presumed aetiology | paramedian pons bleeding possibly due to neurosarcoidosis/pons cavernoma | genetic, neurodegenerative? | neurodegenerative | neurodegenerative |
| therapy | prednisolone, azathioprine for years, 2 cycles rituximab without neurologic improvement | | levodopa treatment over 3 months without significant improvement 4-Aminopyridin for 4 months: no significant improvement in gait | prednisolone, intravenous immunoglobulins, gluten-free diet without neurologic improvement levetiracetam did not improve PT |

Abbreviations: AAO: age at onset, AHI: apnoea–hypopnoea-index, CSF: cerebrospinal fluid, DD: disease duration, DM: diabetes mellitus, ESS: Epworth sleepiness scale, FSS: Fatigue severity scale, HOD: hypertrophic olivary degeneration, HSP7: hereditary spastic paraplegia type 7, Ig: Immunoglobuline, MoCA: Montreal Cognitive Assessment, NBIA: Neurodegeneration with Brain Iron Accumulation, PAPT: progressive ataxia and palatal tremor, PAS: Periodic acid–Schiff, PLM: periodic limb movements, POLG: polymerase gamma gene catalytic subunit, PSG: polysomnography, PT: palatal tremor, RBD: REM sleep-behavior disorder, SWI: susceptibility weighted imaging.

### 3.2. Patient 2

This female patient in her early 60s presented in our clinic with progressive gait ataxia with substantial worsening over the following years, which recently made wheelchair use necessary. She first noticed gait disturbances 20 years prior in her early 40s; years later, limb ataxia developed. Moreover, she complained of difficulties with visual fixation. Past medical history was significant for diabetes mellitus type 2 (diagnosis 14 years prior), Hashimoto's thyroiditis (diagnosis 3 years prior), multiple depressive episodes, and sleep-associated breathing disturbance.

On a subsequent consultation, a low-frequency palatal tremor (no ear click sounds), slight dysphagia, and dysarthria were noted. A video-oculography revealed a fixation deficit, macro square wave jerks and dysmetric saccades (that could at least partially explain the aforementioned problems with visual fixation). An electroneurography showed marked sensory axonal polyneuropathy of the lower extremities. Furthermore, on otologic examination, bilateral sensorineural hearing loss with normal vestibular function was diagnosed. However, hearing difficulties have already been present for years. At most, she showed mild cognitive impairment.

Blood and CSF analyses were normal, and two MRI examinations of the brain showed bilateral HOD and progressive cerebellar atrophy (Figure 1(2A–2D)).

The patient complained of constant fatigue (fatigue severity scale (FSS) 63/63 points) and daytime sleepiness (ESS 12–13/24 points). A multiple sleep latency test (MSLT) was normal. Multiple V-PSG examinations showed pronounced fragmentation of sleep with virtually no REM sleep due to severe central sleep apnoea (apnoea –hyponoea index (AHI) = 60.3/h, lowest blood oxygen concentration 86%, Table 1). AHI improved under continuous positive airway-pressure therapy, but daytime sleepiness and fatigue did not.We suggested genetic testing, which the patient politely declined.

### 3.3. Patient 3

This male patient in his mid-60s presented with a 5 year history of progressive gait ataxia. Past medical history comprised an abdominal aortic aneurysm, arterial hypertension, and extensive nicotine consumption. On examination, he presented with gait and limb ataxia, saccadic eye movements, dysarthria, dysphagia, and a low-frequency palatal tremor of 2 Hz with synchronous, rhythmic click sounds. He showed no rigidity, spasticity, or dystonia. However, the reflex level was elevated, and the arm swing was reduced on both sides. He showed mild cognitive impairment.

Serum examinations revealed a markedly elevated ferritin level (>1000 ug/L, normal <250). CSF, including the search for Whipple's disease, was unremarkable. No other organ system involvement was present; liver pathology (especially haemochromatosis) and systemic inflammation were excluded. Furthermore, the search for malignant tumours was negative.

An MRI of the brain 5 years after symptom onset showed cerebellar and supratentorial brain atrophy, bilateral HOD, and bilateral susceptibility artefacts in the posterior and lateral putamen, red nucleus, and dentate nucleus (Figure 1(3A–3E)).

The patient complained of some daytime sleepiness (ESS 6/24 points), and infrequent snoring was reported with an AHI of 21.1/h on an ApnoeLink screening test (Table 1). There was no history of RBD. On the suspicion of a sleep-associated breathing disturbance, we suggested a V-PSG, which the patient politely declined.

On follow-up examination one year later, the HOD subsided. A treatment trial with levodopa (for 3 months) and 4-aminopyridine (for 4 months) unfortunately showed no significant neurologic improvement.

With the progression of the abdominal aortic aneurysm with an increasing likelihood of rupture, the patient wished to continue follow-ups nearer to his hometown and did not wish for further diagnostic and therapeutic interventions.

### 3.4. Patient 4

This male patient presented with a 3 year history of gait disturbances, dysarthria, and dysphagia, with onset in his early 50s. The past medical history comprised only of latent pulmonary tuberculosis. On examination, he showed hypometric saccades and rotatory nystagmus, palatal tremor of approximately 3–5 Hz, and bulbar symptoms, as well as marked gait, trunk, and limb ataxia, proximal spastic paresis, and symmetrical pallhypesthesia of the lower extremities that made him almost exclusively wheelchair bound.

Over the course of one year, the bulbar symptoms markedly worsened with distinct dysphagia and an intermittent stridor when breathing. He developed myoclonus in the periorbital, periocular, and neck regions that was synchronous with the palatal tremor (3–5 Hz on surface electromyography), showing resemblance to the myokymic discharges described by Sidiropoulos et al. [23]. Furthermore, he developed urinary urgency and complained of dizziness at orthostatic positional changes.

Multiple MRI examinations of the brain showed progressive cerebellar and marked brainstem and cervical myelon atrophy. Two years after symptom onset, subsequent bilateral HOD was depicted (Figure 1(4A–4D)).

Laboratory workup showed slightly elevated gliadin immunoglobulin A. There were no signs of Wilson's disease or Niemann–Pick Disease. Three lumbar punctures within one year revealed slightly elevated cell counts (between five and eight cells/µL), with slightly elevated protein in the first puncture. Autoimmune and paraneoplastic antibodies were normal (in serum and CSF). The search for Whipple's disease and active mycobacterial disease was negative. Extensive tumour search, including whole body CT, was inconspicuous.

Temporarily, he manifested vivid hallucinations of people during the night and sleep disturbances without vivid dreams. He complained of worsening daytime sleepiness (ESS 10/24 points) and general fatigue. On V-PSG, he showed a drastically reduced sleep latency (patient fell asleep during the preparation phase), severe obstructive sleep apnoea (AHI 71.6/h, hypercapnia with a mean $CO_2$ of 54 mmHg), and periodic limb movements without REM/non-REM sleep-behaviour disorder (Table 1).

Due to the rapid progression of the bulbar symptoms and because no evidence of an infectious agent was present, the patient underwent treatment with high-dose methylprednisolone, followed by intravenous immunoglobulins, without any clinical improvement. A strict gluten-free diet did not show any therapeutic response. Recently, at a different hospital nearer to his hometown, a genetic panel diagnostic for ataxia and spastic paraparesis-associated genes was initiated, which was unremarkable (amongst other genes: SCA 1, 2, 3, 6, 7, 17, Friedreich-ataxia, FXTAS, CYP27A1, GFAP, HEXA/HEXB, NPC1/2, SPG7, POLG, Table A1).

## 4. Discussion

The aim of our study was to highlight the broad clinical spectrum and the heterogeneous aetiology of PAPT syndromes and estimate the prevalence of this rare disease (estimated between 0.26 and 1.05 per 100.000). To our knowledge, we are the first to report on sleep studies in presumed degenerative PAPT syndrome.

Since the publication of the first large PAPT cohort in 2004 [6], many new case reports and small case series surfaced. In addition to the progressive cerebellar syndrome and the palatal tremor, the clinical picture of PAPT syndromes seems to be more heterogeneous than initially described [6]. Pyramidal and extrapyramidal symptoms, neuropathy, bulbar symptoms, hearing loss, cognitive and psychiatric symptoms, and autonomic features, as well as seizures and optic atrophy, can be present [5]. Sleep disturbances were prevalent in three out of our four patients, a finding that was, up until now, rarely reported in PAPT patients (one case report mentions sleep disturbances in a patient with hereditary spastic paraplegia type 7 (HSP7) [22] and one in adult-onset Alexander's disease [24]).

In the following, we will discuss the clinical characteristics, imaging findings, and possible aetiologies of the PAPT syndromes of our patients in light of the current literature. A summary of mean age at onset, gender distribution, and imaging features of PAPT patients found in the literature are assembled in Table 2.

**Table 2.** Summary of PAPT syndrome characteristics in the literature, including our cases.

| PAPT Syndrome | Number of Cases | Age at Onset in Years (Mean $\pm$ Standard Deviation) | Gender (Female) | Cerebellar Atrophy (%, n) | Hypertrophic Olivary Degeneration (%, n) |
|---|---|---|---|---|---|
| idiopathic/neurodegenerative [6,9–11,25–39] | 51 | $54.7 \pm 13.8$ | 24% (12/49) | 79% (31/39) | 86% (32/37) |
| structural (vascular, infectious, autoimmune) [12–16,23,40,41] | 15 | $50.9 \pm 14.4$ | 53% (8/15) | 50% (8/15) | 93% (14/15) |
| genetic/familial (confirmed or suspected) [6,17–22,24,42–46] | 34 | $41.7 \pm 12.2$ | 54% (13/24) | 88% (29/33) | 56% (10/18) |
| | | $p < 0.001$ | $p = 0.02$ | $p = 0.03$ | $p = 0.009$ |

This table compiles the numbers of imaging/pathologically proven cerebellar atrophy and hypertrophic olivary degeneration (when information was available). It shows unadjusted $p$-values for group differences calculated using chi-squared test or ANOVA (analysis of variance).

### 4.1. Patient 1

A single persistent lesion within the Guillain–Mollaret triangle (usually haemorrhage, cavernoma, or vascular malformation) can occasionally lead to PAPT syndrome [12–14]. Pathophysiologically, it is speculated that superficial or intraparenchymal hemosiderin accumulation might lead to iron-associated neurodegeneration [12], leading to progressive symptoms. The static SWI lesion in the pons of this patient could be suggestive of a similar pathomechanism, with the development of a PAPT syndrome after a one-time event.

Although a definitive pathological diagnosis is lacking, the elevated CD4/CD8 cell ratio of 11 in the bronchoalveolar lavage (specificity and positive predictive value almost 100%; lymphomas show significantly lower ratios between 0.9–3.4 [47,48]), the pulmonary findings and the clinical and CSF presentation initially suggested sarcoidosis with neurological manifestations [23]. Neurosarcoidosis can present with T2 hyperintense findings on a brain MRI [49] and rarely can lead to intracranial haemorrhages [50], possibly providing an explanation for the hypointense SWI lesion in the dorsomedial pons in our patient. Nevertheless, we cannot exclude that the dorsomedial pons lesion was caused by an "independent" cavernoma or telangiectasia. It is rather unlikely that the slow progression of the symptoms over years under appropriate immunosuppressive therapy would be attributable to neurosarcoidosis.

This case underscores the importance of the search for haemorrhage, cavernoma, vascular malformation, and iron and hemosiderin deposits, as well as signs of inflammation on a brain MRI. In addition, it should be kept in mind that HOD might not be present on the initial brain MRI and might only develop later or sequentially (as was the case in this patient) and may subside over time (as seen in case 3) [51]. Interestingly, the proportion of patients with cerebellar atrophy seems to be lower in the subgroup of structural PAPT syndrome (Table 2). Furthermore, the search for infectious and autoimmune agents in serum and CSF might provide targets for causal treatment.

*4.2. Patient 2*

This patient presents a complex PAPT syndrome with a combination of cerebellar, oculomotor and psychiatric disease, sensorineural hearing loss, sensory polyneuropathy, mood and sleep disturbance, and fatigue. The disease onset in the early 40s (see Tables 2 and A1) and the sleep disorder that showed central sleep apnoea with an increased arousal index and a significantly decreased percentage of REM sleep could be suggestive of a genetic PAPT syndrome [52]. Although there is considerable phenotypic overlap between the genetic syndromes (Table A1), interestingly, our patient exhibited seven of the clinical signs used to assess the sensitivity/specificity of neurological symptoms in patients with and without (homozygous) POLG mutations (that can exhibit a predominantly or pure PAPT phenotype with HOD on brain imaging [17,18,53,54]), reaching a sensitivity/specificity of over 60–70% [55]. The relatively early presentation of diabetes mellitus could support this diagnosis [56].

Nonetheless, genetic confirmation is lacking and an idiopathic PAPT syndrome or coincidental symptoms are possible.

Although initially HOD was thought to only appear in sporadic PAPT [6], this patient, as well as the literature, indicates that HOD can be present in familial/genetic PAPT syndromes albeit in a lower percentage of cases (Table 2).

Finally, this patient highlights the importance of V-PSG in the search for sleep apnoea, hypoventilation, and (non-)RBD, as well as the implementation of a screening for polyneuropathy and myopathy.

*4.3. Patient 3*

The distinctive features of this patient are the isolated elevation of serum ferritin and the iron deposits in the posterior and lateral putamen, as well as a red and dentate nucleus on a brain MRI (Figure 1(3A–3E)). Neurodegeneration with brain iron accumulation (NBIA) can manifest with movement disorders at an adult age [57]. One case report describes palatal tremor and chorea in a patient with NBIA; however, ataxia was absent [58]. However, the constellation of ferritin, copper, and ceruloplasmin in our patient is not consistent with subtypes of NBIA (neuroferritionopathy: serum ferritin usually decreased or normal; aceruloplasminemia: ceruloplasmin and copper are usually decreased or undetectable [59]). Furthermore, the iron deposits on brain MRIs are normally present in most of the basal ganglia, i.e., globus pallidus, caudate, and putamen [59,60], not just in the lateral and posterior parts of the putamen, as in our patient.

With the lack of other seminal findings on extensive aetiological work-up, the relatively isolated clinical PAPT syndrome, the disease course, and the cerebral imaging in this patient would be consistent with case reports of patients with presumed idiopathic PAPT [6,9–11,25–39].

The recent findings in two pathological studies of three idiopathic PAPT patients point towards a novel neurodegenerative disease with tau pathology [9,10]. It might be conceivable that the iron deposits in our patient are merely an epiphenomenon of an underlying neurodegenerative process, which is similar to that seen in atypical Parkinson's syndromes. Both alpha-synucleinopathy (e.g., multiple system atrophy, MSA) and tauopathy (e.g., progressive supranuclear palsy, PSP) sometimes show relatively specific iron accumulation patterns in the (posterior and lateral) putamen, red nucleus, and dentate nucleus on a brain MRI [61,62].

Usually, the main complaint of patients with PAPT syndrome is attributable to progressive ataxia. In this patient, ear click sounds were present (more frequent in essential PT, but seldom present in symptomatic PT and PAPT [5]), which led to the diagnosis. However, in the absence of ear clicks, the palatal tremor often remains unmentioned by the patient and unnoticed by the physician, albeit it is a seminal clinical finding. In fact, in the first two patients presented here, PT was only discovered in the context of patient rounds by the senior author of this paper. This, on the one hand, emphasises the necessity of a thorough physical examination, and, on the other hand, highlights the fact that one only sees what one looks for (and what one knows).

*4.4. Patient 4*

The slightly elevated cell count and protein in CSF is suggestive of an inflammatory process. Tests for common infectious diseases (e.g., HIV, Syphilis), as well as mycobacteria and Whipple's disease, were negative. Concerning autoimmune aetiologies, multiple cases have been described in the literature with fitting clinical presentations ("chronic lymphocytic inflammation with pontine perivascular enhancement responsive to steroids" CLIPPERS) [63], anti-NMDA receptor [63], anti-GAD65 antibody encephalitis [40], and neurosarcoidosis [23]. In contrast, our patient exhibited no autoimmune antibodies and showed no clinical or radiological improvement to corticosteroid therapy. Furthermore, no signs of sarcoidosis were present on both a brain MRI and a whole-body CT, and, because the patient's condition declined despite adequate immunosuppressive treatment, we found this aetiology sufficiently unlikely.

There are a few case reports describing PAPT syndrome in the context of gluten sensitivity [15,64–66]. Although our patient presented with PAPT syndrome and positive anti-gliadin antibodies, the clinical picture was markedly more complex than that described in case reports; our patient did not show any stagnation or improvement with a strict gluten-free diet.

Recently, the anti-IgLON5 syndrome was described, a novel autoimmune-mediated, neurodegenerative disease, characterized by non-REM and REM parasomnia, sleep apnoea, stridor, bulbar dysfunction, dysautonomia, and, sometimes, movement disorders, with antibodies against the neuronal cell-adhesion protein IgLON5 and tau pathology on histologic examination of the brain [67]. In our patient, IgLON5 antibodies were negative in serum and CSF (also, to our knowledge, palatal tremor was not described in this disease).

The clinical phenotype of our patient could be compatible with the complicated phenotypes of many genetic PAPT syndromes that are summarized in Table A1. Although initially hesitant, the patient recently underwent a genetic panel testing that excluded most genetic diseases except for spinocerebellar ataxia type 20 which, however, does not fit the clinical phenotype [19].

In conclusion, the aetiology of this PAPT syndrome is most likely idiopathic. Nonetheless, mutations not registered via the used sequencing method would still be possible (e.g., expansion mutations, mitochondrial DNA).

Finally, the possibility of a cerebellar subtype of multiple system atrophy (MSA) needs to be mentioned because of the prominent cerebellar and pyramidal signs, the dysautonomic features, and the sleep disturbances [68]. As mentioned elsewhere [6], up until this point, PT was not described in pathologically confirmed MSA patients. Moreover, in an extensive review of HOD aetiologies, the study did not find any patients with MSA as the underlying cause [69]. However, the fact that PT was described in patients with progressive supranuclear palsy (PSP) [39,70] would be in keeping with the presumed tau pathology of idiopathic PAPT syndromes [9–11].

A possible diagnostic approach to PAPT syndromes is compiled in Table 3.

**Table 3.** Diagnostic approach.

| | PAPT Syndrome | Additional Assessments in PT/Ataxia |
|---|---|---|
| serum | Anti-gliadin antibodies (gluten sensitivity) [15]<br>ACE (Sarcoidosis) [23]<br>Copper/coeruloplasmin (Wilson's disease) [71]<br>Ferritin (NBIA) [58]<br>Cholesterine, cholestanol, bile alcohol (CTX) [46]<br>Autoimmune encephalitis antibodies<br>(anti-NMDA-receptor, IgLON5, Anti-GAD65 [40]<br>if appropriate consider panel diagnostic) [63,67]<br>If appropriate: hexosaminidase enzymatic activity in leukocytes and serum [45] | Infectious disease (HIV, Syphilis, Lyme disease,<br>Vitamins E and B12, folic acid<br>Anti-TG/-TPO antibodies (Hashimoto thyroiditis)<br>ANA, anti-SSA(Ro) (connective tissue disease)<br>Paraneoplastic antibodies (if appropriate consider panel diagnostic: Anti-Hu, Yo, Ri, Tr, ZIC4, GAD, CRMP-5, Ma1, Ma2, Amphiphysin)<br>If appropriate: long-chain fatty acids |
| urine | Copper in 24 h urine (Wilson's disease) [71] | |
| CSF | Cell count, protein (infectious/autoimmune),<br>ACE (Sarcoidosis) [23] | Oligoclonal bands (demyelinating disease)<br>Malignant cells<br>PCR for Whipple's disease<br>If appropriate: Mycobacterial culture |
| MRI | MRI of brain and spine:<br>Atrophy of cerebellum, medulla, cervical spine (tadpole pattern of atrophy [72])<br>HOD (unilateral, bilateral) [69]<br>Haemorrhage, cavernoma, vascular malformation, tumour especially in pons, midbrain, cerebellum [13]<br>Superficial hemosiderosis, iron deposits? [12,60]<br>Calcifications (SCA20, pons calcification [14]) | |
| electrophysiological study | Polyneuropathy screening<br>Muscle atrophies, myopathy<br>Surface EMG (frequency of PT)<br>If appropriate: audiometry | EEG (seizure) |
| genetic study | Consider panel analysis, including mutations in:<br>GFAP (Alexander's disease) [20,21], HSP7 [22], SCA20 [19], Mitochondrial diseases (POLG, SURF1) [17,18,53], HEXA/HEXB (GM2-gangliosidosis) [45], CYP27A1 (CTX) [46], Ferritin light chain [58] | Consider genetic panel analysis |
| other | Video-polysomnography (NREM/REM sleep, hypoventilation, sleep apnoea)<br>Screening for M. Behçet [41]<br>If appropriate: video-oculography<br>if appropriate: CT/PET-CT of chest, abdomen: tumour, sarcoidosis, other organ involvement | If appropriate: Dopamine-Transporter Scan (degenerative parkinsonism)<br>Extended tumour search |

Abbreviations: ACE: angiotensin-converting enzyme, CSF: cerebrospinal fluid, CTX: Cerebrotendinous xanthomatosis, EEG: electroencephalogram, EMG: electromyography, HOD: hypertrophic olivary degeneration, NBIA: Neurodegeneration with brain iron accumulation, NREM: nonrapid eye movement, REM: rapid eye movement.

## 5. Conclusions

Our study highlights the diverse clinical spectrum and the heterogeneous aetiology of PAPT syndromes and provides a local prevalence estimate. Despite the syndromal overlap of the aetiologic subgroups, they seem to differ in age at onset, gender distribution and frequency of cerebellar atrophy, and hypertrophic olivary degeneration.

Three out of our four patients reported daytime sleepiness and showed sleep-associated breathing disturbance; two of the patients had affected sleep architecture, and one had significantly reduced sleep latency. Only the first patient with the presumed "structural" aetiology of paramedian pons bleeding (and no underlying "neurodegenerative" disease per se) did not exhibit sleep disturbances. In aggregate, we provide evidence that sleep disturbances seem to be part of the PAPT syndrome. Analogous to other tau pathologies (e.g., PSP) [73], RBD seems not to be a dominant finding, at least in our small cohort.

Our study further emphasises the importance of palatal inspection for it bears the potential of significantly narrowing down the list of differential diagnoses and the need for a proactive search of potentially causally treatable diseases that could change the disease course (remarkably).

Further investigations are required to gain more understanding of the motor and especially non-motor symptoms, the genetic landscape, and the pathophysiological mechanisms of this disease to be able to make sounder differential diagnoses and, in the future, potentially provide more targeted/individualised therapies.

**Author Contributions:** Conceptualization, N.S. and C.L.A.B.; literature research, selection of literature, writing—original draft preparation, N.S.; interpretation of MRI findings, R.W.; writing—review and editing, N.S., R.W. and C.L.A.B.; supervision, C.L.A.B. All authors have read and agreed to the published version of the manuscript.

**Funding:** This research received no external funding.

**Institutional Review Board Statement:** The study was conducted in accordance with the Declaration of Helsinki. According to the guidelines of the Ethics Committee of the Canton of Bern, no separate ethical approval for case reports is necessary if the patient has given informed consent. No animals and no further human data or tissues were studied.

**Informed Consent Statement:** Informed consent was obtained from all patients involved in the study.

**Data Availability Statement:** The data presented in this study are available on request from the corresponding author. The data are not publicly available due to their containing information that could compromise the privacy of the research participants.

**Acknowledgments:** We kindly thank the patients for allowing the publication of their medical histories.

**Conflicts of Interest:** The authors declare no conflict of interest.

## Appendix A

**Table A1.** Summary of genetic PAPT syndromes.

| | HSP7 [22,74] | Adult-Onset Alexander's Disease [21,22, 42,43,72,75] | Spinocerebellar Ataxia 20 [19,44] | POLG [55,76] | Neuroferritinopathy/ Aceruloplasminemia [57,59] | Late-Onset GM2-Gangliosidosis [45,77–79] | CTX [80] |
|---|---|---|---|---|---|---|---|
| Age at onset | 11–72 | 13–62 | 19–64 | adolescent/adult | early to mid-adulthood | 10–54 | 23–44 |
| Oculomotor symptoms | ptosis nystagmus ophthalmoplegia | ptosis nystagmus diplopia | dysmetric saccades | ptosis external ophthalmoplegia | blepharospasm in aceruloplaminemia | dysmetric saccades | – |
| Bulbar symptoms | + | ++ | dysarthria, dysphonia | + | +/− | + | +/− |

**Table A1.** *Cont.*

| | HSP7 [22,74] | Adult-Onset Alexander's Disease [21,22, 42,43,72,75] | Spinocerebellar Ataxia 20 [19,44] | POLG [55,76] | Neuroferritinopathy/ Aceruloplasminemia [57,59] | Late-Onset GM2- Gangliosidosis [45,77–79] | CTX [80] |
|---|---|---|---|---|---|---|---|
| PT | + | + | + | + | + | + | + |
| Hearing loss | + | − | − | + | − | − | − |
| Gait abn. | +++ | ++ | + | + | +/− | ++ | + |
| Axial ataxia | + | + | + | + | + | + | + |
| Limb ataxia | + | + | + | + | + | + | + |
| Spasticity | +++ | ++ | +/− | − | − | +/− | + |
| Paresis | ++ | + | − | limb weakness | − | ++ | + |
| Babinski sign | ++ | + | − | − | − | ++ | + |
| Amyotrophy | + | + | − | − | − | ++ | +/− (myopathy) |
| PNP | + | +/− | − | + | − | + | + |
| Other typ. symptoms | optic atrophy | − | tremor | myoclonus epilepsy movement disorders, multiorgan involvement possible | movement disorders (chorea, dystonia, facial dyskinesia) multiorgan involvement | movement disorders (dystonia) | cataract, tendon xanthomas epilepsy, parkinsonism, multiorgan involvement |
| Cognitive abn. | +/− | − | − | + | ++ | + | + |
| Psychiatric disease | − | − | − | + | + | + | + |
| Sleep disturbance | + | + | − | + | − | + | − |
| Scoliosis | + | +/− | − | − | − | − | − |
| Sphincter abn. | + | + | − | − | − | +/− | +/− |
| Dysautonomy | − | + | − | − | − | +/− | +/− |
| MRI findings | cerebellar atrophy, white matter abnormalities | atrophy of medulla oblongata and cervical spine ("tadpole pattern of atrophy" [42,72]) HOD, supratentorial white matter abnormalities | HOD, isolated dentate calcification | mild cerebellar atrophy, HOD | iron deposits in iron-rich brain regions: basal ganglia, thalamus, dentate nucleus, substancia nigra, cerebral atrophy | cerebellar atrophy, hypodensity of thalamus | cerebellar atrophy, white matter signal alterations, symmetric hyperintensities in the dentate nuclei |

Abbreviations: abn.: abnormalities, CTX: Cerebrotendinous xanthomatosis, HSP7: hereditary spastic paraplegia type 7, PNP: polyneuropathy, POLG: polymerase gamma gene-catalytic subunit.

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
