# Peer review of "Four New Cases of Progressive Ataxia and Palatal Tremor (PAPT) and a Literature Review"

_ctn, doi:10.3390/ctn7040032_

Round 1
Reviewer 1 Report
1. How do the authors know that are less than 100 cases of ataxia and palatal tremor in the literature? It is better to remove these specific numbers because it was not the aim of the manuscript.
2. Figure 1. “1c.” “3B” and “3C” Are the arrow in the correct position?
3. The authors should change the manuscript, suggesting that the case they are describing is possibly Progressive Ataxia and Palatal Tremor. The description of genetic studies should always support the diagnosis of this uncommon syndrome. There is a significant overlap of symptoms with many other conditions that already have genetic analysis.
Reviewer 2 Report
The authors present a series of 4 cases with palatal tremor and progressive ataxia, in addition to a short overview of 96 similar cases collected from the literature. Gait and limb ataxia were both observed in their own cases. They present the differential diagnosis of their cases at length, provide information on (minor) sleep disturbances, suggest a diagnostic work-up in similar cases, and provide a table on rare genetic causes of the syndrome. They make clear that both the range of associated symptoms and the range of underlying conditions is broad.
They claim that the observation of palatal tremor narrows the list of conditions to be considered (l. 371; although it still remains remarkably diverse) and that correct diagnosis offers potential therapeutic benefit (l. 380; although none of their cases responded to treatment). Immune therapy was without success in two of their own cases.
On the whole, their presentation is thorough and balanced.
Some minor points remain:
The presumed diagnosis of POLG disease in case 2 (table 1) sounds unconvincing to me.
The presumed diagnosis of case 3 is neurodegenerative (table 1). Why has hypertrophic olivary degeneration subsided then?
The authors do not mention Holmes tremor (which has a cerebellar ataxia component) which can accompany palatal tremor: Are there no cases combining these features? Can we be sure that both features have been safely distinguished in published cases?
I doubt that facial myokymia (case 4, l. 170) has been confirmed electrodiagnostically; similar cases have been diagnosed as myoclonus according to electrophysiological examination.
Why was levodopa tried in case 3 (and not, for example, lamotrigine, valproic acid or botulinum toxin, that in some cases have improved palatal tremor)?
If tau has something to do with palatal tremor (l. 292), have tau levels been examined in CSF?
Of the 100 cases described to date, how many of them have cerebellar atrophy and how many have hypertrophic olivary degeneration?
If it is adequate to group all the diverse cases displaying palatal tremor and progressive ataxia under one heading, what is the presumed neuroanatomical substrate? What distinguishes these cases from those with palatal tremor alone?
The symptoms of the 4 own cases started in middle age: What is the range of age at the onset of symptoms in other published cases?
Some minor language items remain (e.g. l. 33 plurality (better: number or plethora); l. 121 expansive (better: extensive); l.379 dominating (better: dominant); aceruloplasminemia (appendix). We would prefer to use the term spinocerebellar ataxia for SCA (l. 409).
